# Prevalence of TB symptoms, diagnosis and treatment among people living with HIV (PLHIV) not on ART presenting at outpatient clinics in South Africa and Kenya: baseline results from a clinical trial

Alana Brennan [1,2,3] Mhairi Maskew,[2] Bruce A Larson [3] Isaac Tsikhutsu,[4,5] Margaret Bii,[4,5] Lungisile Vezi,[2] Matthew Fox,[1,2,3] Willem Daniel Francois Venter [6] Peter D Ehrenkranz,[7] Sydney Rosen [2,3]

For numbered affiliations see end of article.

**Correspondence to**
Alana Brennan;
abrennan@bu.edu

## ABSTRACT

**Objective** We used screening data and routine clinic records for intervention arm patients in the Simplified Algorithm for Treatment Eligibility (SLATE) trials to describe the prevalence of tuberculosis (TB) symptoms, diagnosis and treatment among people living with HIV (PLHIV), not on antiretroviral therapy (ART) and presenting at outpatient clinics in South Africa and Kenya. We compared the performance of the WHO four-symptom TB screening tool with a baseline Xpert test.

**Setting** Outpatient HIV clinics in South Africa and Kenya.

**Participants** Eligible patients were non-pregnant, PLHIV, >18 years of age, not on ART, willing to provide written informed consent. A total of 594 patients in South Africa and 240 in Kenya were eligible.

**Results** Prevalence of any TB symptom was 38% in Kenya, 35% (SLATE I) and 47% (SLATE II) in South Africa. During SLATE I, 70% of patients in Kenya and 57% in South Africa with ≥1 TB symptom were tested for TB. In SLATE II, 79% of patients with ≥1 TB symptom were tested. Of those, 19% tested positive for TB in Kenya, 15% (SLATE I) and 5% (SLATE II) tested positive in South Africa. Of the 28 patients who tested positive in both trials, 20 initiated TB treatment. The lowest median CD4 counts were among those with active TB (Kenya 124 cells/mm$^3$; South Africa 193 cells/mm$^3$). When comparing the WHO four-symptom screening tool to the Xpert test (SLATE II), we found that increasing the number of symptoms required for a positive screen from one to three or four decreased sensitivity but increased the positive predictive value to >30%.

**Conclusions** 80% of patients assessed for ART initiation presented with ≥1 TB symptoms. Reconsideration of the 'any symptom' rule may be appropriate, with ART initiation among patients with fewer/milder symptoms commencing while TB test results are pending.

**Trial registration number** NCT02891135 and NCT03315013.

## Strengths and limitations of this study

► The data for the analysis presented here come from the intervention arms of two randomised clinical trials (Simplified Algorithm for Treatment Eligibility (SLATE) I in South Africa and Kenya and SLATE II in South Africa), which were conducted at typical primary healthcare clinics in South Africa and typical hospital-based HIV clinics in Kenya.

► The studies enrolled adults not yet on antiretroviral therapy (ART), including those just diagnosed and those who had already received some pre-ART care, and collected data at study enrolment about tuberculosis (TB) symptoms, TB testing and TB test results.

► We describe types and numbers of symptoms among those with and without TB and compare the performance of symptom screening to laboratory TB test results.

► Limitations of the analysis include heavy reliance on routinely collected data of TB tests performed and their results, likely leading to some missing and incomplete data, and the geographic clustering of the sites in each country, limiting geographic generalisability.

## INTRODUCTION

In 2017, the WHO began recommending rapid antiretroviral therapy (ART) initiation, including same-day initiation (SDI) of ART, after the results of several studies indicated that it could reduce loss to follow-up in the pre-ART period.[1–3] The possibility of coinfection with tuberculosis (TB), however, remains a major reason for delaying ART among those with the TB symptoms on the WHO four-symptom TB screen (cough, weight loss, fever, and night sweats). This symptom screen has been shown to have good sensitivity (89%) but poor specificity (28%) in ART naïve people living with HIV (PLHIV).[4] According to the WHO and national guidelines in both

South Africa[5] and Kenya,[6] patients who report ≥1 TB symptoms require further investigation for active TB disease before ART initiation, which entails a laboratory test such as Xpert mycobacterium TB complex/rifampin (MTB/RIF). Following the TB test, patients with negative results resume regular procedures for ART initiation, while those found to have TB are started on TB therapy, with ART initiation delayed until patients are regarded as stable on TB treatment. As a second clinic visit is typically required to receive TB test results, SDI may be impossible for patients presenting with TB symptoms.

The Simplified Algorithm for Treatment Eligibility (SLATE) I study in South Africa and Kenya[7] and the SLATE II study in South Africa[8] evaluated a clinical algorithm to assess eligibility for rapid ART initiation in patients presenting for HIV care but not currently on ART. The algorithms distinguished between patients eligible for SDI of ART and patients requiring referral to clinic staff for additional standard of care evaluation and TB treatment before ART initiation. One or more symptoms of TB—cough, fever, weight loss or night sweats—of any duration or intensity were among the criteria for referral in SLATE I. SLATE II revised the SLATE I algorithm to allow patients with mild TB symptoms and a negative lipoarabinomannan (LAM) assay test[9] to initiate on the same day, based on clinician judgement.

The purpose of this analysis is to describe the prevalence of TB symptoms among patients presenting for HIV care but not currently on ART and to estimate the rates of TB testing, diagnosis and treatment using baseline screening data and routine clinic records for intervention arm patients in the SLATE I and II clinical trials in South Africa and Kenya. We also assessed the performance of the WHO four-symptom TB screening tool compared with Xpert MTB/RIF sputum testing. The implications of current TB screening, diagnosis and treatment for implementation of SDI are discussed.

## METHODS
### SLATE trials
The SLATE I and SLATE II trials in South Africa and Kenya were multicentre trials evaluating two variations of a simplified algorithm to determine eligibility for SDI of ART without relying on laboratory results or multiple clinic visits.[7 8] Enrolment for SLATE I (NCT02891135) was completed in July 2017 in South Africa and April 2018 in Kenya. Enrolment for SLATE II (NCT03315013), only conducted in South Africa, was completed in September 2018.

### Study population
Patient inclusion and exclusion criteria for study eligibility have been described previously.[7 8] Briefly, those enrolled were non-pregnant, PLHIV, >18 years of age, not currently on ART and willing to provide written informed consent, who were randomised 1:1 to the intervention and standard of care arms. This analysis is limited to patients randomised to the intervention arm, for whom we have TB symptom data. Symptom screening of patients in the standard of care arm was poorly documented by clinic staff. Intervention arm patients were assessed for eligibility for SDI by a study nurse (South Africa) or clinical officer (Kenya) using the SLATE I or II algorithm, which each consisted of four screens: (1) current symptoms, (2) recent medical history, (3) physical conditions and (4) treatment readiness (see online supplementary figures 1,2). The baseline characteristics of these patients have previously been described.[10]

### Patient and public involvement
Patients were not involved in the design, recruitment or conduct of the SLATE trials. We conducted a qualitative study at the end of SLATE II to help gain a deeper understanding of patient perceptions of initiating ART per standard of care compared with SDI using the SLATE algorithm. Presentation of primary study results will be conducted by study staff at participating clinics to disseminate research findings to staff and patients prior to funding ending in July 2020.

### TB screening and diagnosis differences between SLATE I and SLATE II
The main difference between the algorithm used in SLATE I and the revised version used in SLATE II was the approach to TB screening and diagnosis. Both algorithms asked patients for self-reported TB symptoms (any cough, fever, weight loss or night sweats of any duration), based on the WHO four-symptom screen.[11] Procedures differed from that point forward, as described below and illustrated in online supplementary figures 1,2.

In SLATE I, consistent with national guidelines in both countries, patients reporting any symptoms of TB of any severity or duration were referred to routine clinic care, which should have included a TB test according to guidelines. Each patient was given a referral letter for the clinic indicating the reason for referral, such as the presence of TB symptoms, but no further effort was made by study staff to ensure that the patient remained in care or was tested or treated for TB.

As has previously been reported,[12] in SLATE I and illustrated in figures 1, 2, a larger proportion of intervention arm patients than expected were ineligible for SDI due to TB symptoms (38% in Kenya and 37% in South Africa). The presence of ≥1 more TB symptoms was by far the main reason for ineligibility under the SLATE I algorithm. Very few of these patients were ultimately confirmed to have TB, however, and study patients reported no TB-related adverse events after starting ART. We thus speculated that referring a patient out for mild TB symptoms, without further complications, was too stringent a requirement, and we developed the SLATE II algorithm accordingly.

In SLATE II, if a patient reported ≥1 TB symptoms, the newly developed TB module in the algorithm was applied. The TB module included: (1) a more detailed medical history (eg, inquiring about severity and duration of

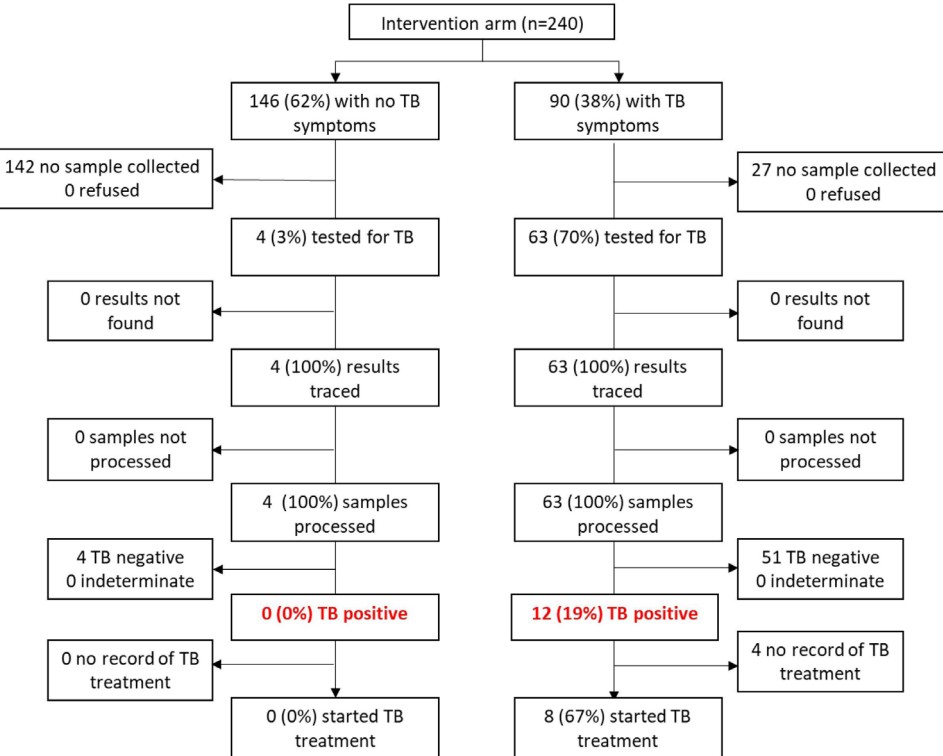

**Figure 1** TB testing flowchart at study enrolment among 240 SLATE I intervention arm participants Kenya. SLATE, Simplified Algorithm for Treatment Eligibility; TB, tuberculosis.

symptoms), (2) a focused physical examination to assess TB symptoms, (3) sputum collection for Xpert MTB/RIF sputum testing and (4) a point-of-care, urine-based LAM test (Determine TB LAM Ag, Abbott, Waltham,

Massachusetts, USA).[9] Patients with a positive LAM test, TB symptoms that were severe or of long duration or any other clinical finding indicating active TB were referred back to routine care under the SLATE II algorithm. As

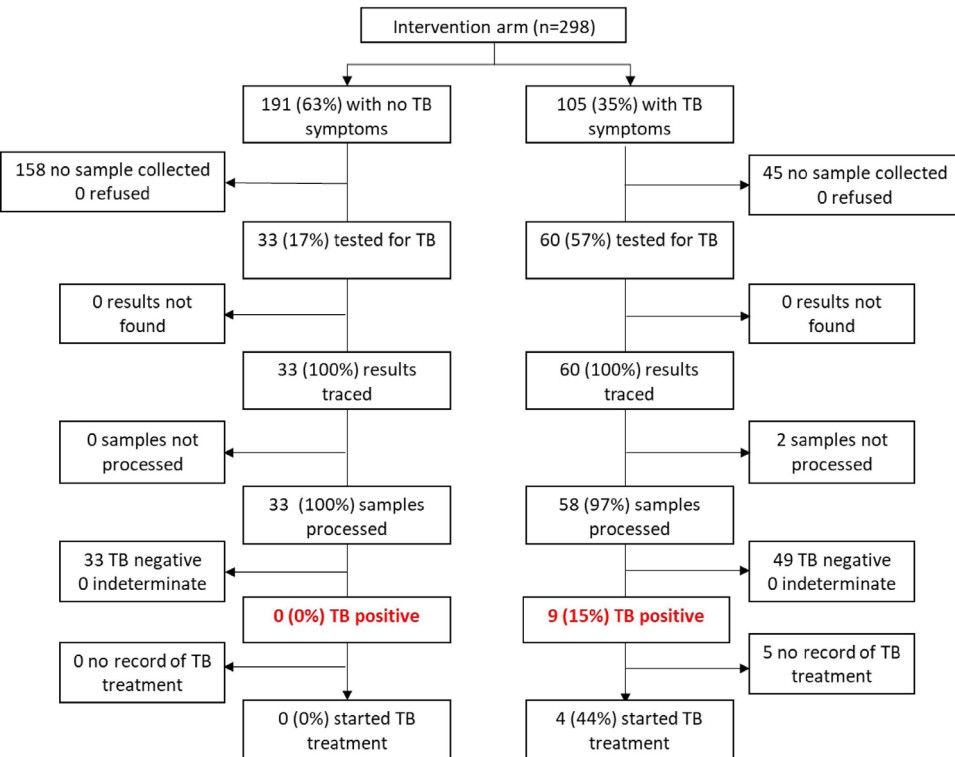

**Figure 2** TB testing flowchart at study enrolment among 298 SLATE I intervention arm participants South Africa. SLATE, Simplified Algorithm for Treatment Eligibility; TB, tuberculosis.

with SLATE I, no effort was made by study staff to ensure that the patient underwent a TB test once referred back to routine care. A patient with a negative LAM test and without severe symptoms of TB remained eligible for same day initiation of ART. For SLATE II, unlike SLATE I, the study staff collected a sputum sample from all intervention arm patients who were able to produce one, regardless of TB symptoms. These samples were sent to the National Health Laboratory Service for Xpert MTB/RIF sputum testing for TB disease.

### Data collection

We collected demographic and algorithm data for intervention arm patients via a case report form administered at study enrolment. Laboratory tests results (eg, CD4 counts and Xpert MTB/RIF findings) were extracted directly from laboratory electronic records or paper-based registers kept at each site, while follow-up data for the study period were collected from routinely generated clinical record data from patient records in electronic and paper format.

### Statistical analysis

Simple descriptive statistics were used to display demographic and clinical characteristics of patients at study enrolment, prevalence of TB symptoms, TB diagnosis and TB treatment.

Having Xpert MTB/RIF sputum tests on the majority (n=214) of SLATE II intervention arm patients provided a unique opportunity to assess the performance of the WHO four-symptom TB screening tool. We defined 11 possible interpretations of TB symptom screening results and compared them to the gold standard of Xpert MTB/RIF: (1) any TB symptom, (2) >2 symptoms, (3) >3 TB symptoms, (4) all four TB symptoms, (5) cough alone, (6) fever alone, (7) weight loss alone, (8) night sweats alone, (9) cough and fever, (10) cough and night sweats and (11) cough and weight loss. We calculated the sensitivity (probability of screening positive (using each definition above) when TB disease is present as defined by Xpert MTB/RIF), specificity (probability of screening negative when TB disease is not present as defined by Xpert MTB/RIF), positive predictive value (probability of a patient having TB disease when the screen is positive) and negative predictive value (the probability of a patient not having TB disease when the screen is negative).

## RESULTS
### Demographic and clinical characteristics

Online supplementary table 1 summarises the basic characteristics of intervention arm patients stratified by trial cohort (n=240 for SLATE I Kenya; n=298 for SLATE I South Africa; n=296 for SLATE II). The majority of patients in each group were women in their mid-30s, with a median baseline CD4 count between 272 cells/mm³ and 294 cells/mm³. We excluded four patients in Kenya and one in South Africa in the SLATE I trial who

were known to be on TB treatment at study enrolment (table 1). Further details of the baseline characteristics of the cohorts have been reported previously.[10]

### TB symptom prevalence

The prevalence of any TB symptoms was 38% (95% CI 32% to 44%) in SLATE I in Kenya, 35% (30%–41%) in SLATE I in South Africa and 47% (42%–53%) in SLATE II (table 1). In both studies and both countries, among people with any symptom, cough (66% combined) and weight loss (72% combined) were the most common symptoms reported. As presented in online supplementary table 2, patients with TB symptoms had substantially lower CD4 counts in all three cohorts at study enrolment than did those with no symptoms of TB, indicating more advanced HIV disease among symptomatic patients. We saw little variation in CD4 count when the data were stratified by number of symptoms or symptom type. The lowest median CD4 counts were recorded among those found to have active TB disease (Kenya 124 cells/mm³ (12–150); South Africa (SLATE I and II combined) 193 cells/mm³ (56–223)).

### TB testing and diagnosis among symptomatic patients

In Kenya (SLATE I), 90 (38% (32%–44%)) intervention arm patients had ≥1 symptom of TB and screened out of the algorithm, with referral back to standard care. Clinic staff chose or were able to test only 63/90 (70% (60%–79%)) symptomatic patients for TB. Of those tested, 12/63 (19% (11%–30%)) had a positive result, corresponding to an estimated TB prevalence in symptomatic patients presenting for care in Kenya of 5% (3%–8%) (assuming those not tested were negative for TB disease) (figure 1). In SLATE I in South Africa, we saw a similar proportion of patients presenting with at least 1 symptom of TB (105, 35% (30%–41%)), but only 60 (57% (48%–66%)) of these symptomatic patients were tested for TB by the study clinics. Of those tested, 9/60 (15% (8%–26%)) had a positive result, corresponding to an estimated TB prevalence in patients presenting for care in South Africa during SLATE I of 3% (2%–5%) (figure 2), with the same assumption regarding the negative status of those not tested.

In SLATE II, all 296 intervention arm patients were asked for a sputum sample per study protocol,[8] regardless of symptoms. We were able to successfully collect and test sputum for 111 (79% (72%–85%)) symptomatic patients and 118 (76% (68%–82%)) asymptomatic patients, or 72% (n=214/296) overall (figure 3). Of the 29 symptomatic patients not tested, 25 were unable to produce a sputum sample and 4 refused to test. Among the 106 symptomatic patients in SLATE II with a successful test, 6 (5% (2%–11%)) results were positive for TB, producing an estimated TB prevalence in all PLHIV (asymptomatic and symptomatic) presenting for care in South Africa during SLATE II of 2% (1%–5%), with the same assumption regarding the negative status of those not tested.

**Table 1** Self-reported TB symptoms, TB diagnosis and TB treatment uptake among patients who screened out due to TB symptoms in the intervention arms of the SLATE I and SLATE II trials

| Variable (% responding yes) | Kenya (SLATE I) (N=240) n (%) | South Africa (SLATE I) (N=298) n (%) | South Africa (SLATE II) (N=296) n (%) |
|---|---|---|---|
| Screened for TB | 240 (100) | 298 (100) | 296 (100) |
| Currently on TB treatment* | 4 (2) | 2 (1) | 0 (0) |
| One or more TB symptoms | 90 (38) | 105 (35) | 140 (47) |
| Symptoms reported (among patients with one or more symptoms) | | | |
| Cough (current) | 75 (83) | 71 (68) | 76 (54) |
| Fever | 53 (59) | 45 (43) | 20 (14) |
| Night sweats | 56 (62) | 44 (42) | 29 (21) |
| Weight loss | 72 (80) | 76 (72) | 96 (69) |
| Number of symptoms reported (n) | | | |
| One symptom | 13 (14) | 36 (34) | 89 (64) |
| Two symptoms | 18 (20) | 27 (26) | 32 (23) |
| Three symptoms | 29 (32) | 22 (21) | 8 (6) |
| Four symptoms | 30 (33) | 20 (19) | 11 (8) |
| TB test performed in symptomatic patients | 63 (70) | 60 (57) | 111 (79) |
| Positive TB tests among symptomatic patients | 12 (13) | 9 (9) | 6 (5) |
| Symptoms among those testing positive | | | |
| Cough (current) | 12 (100) | 9 (100) | 6 (86) |
| Fever | 9 (75) | 7 (78) | 4 (57) |
| Night sweats | 12 (100) | 7 (78) | 4 (57) |
| Weight loss | 10 (83) | 9 (100) | 5 (71) |

*Excluded from analyses.
SLATE, Simplified Algorithm for Treatment Eligibility; TB, tuberculosis.

Among the total of 28 TB-positive patients in both studies, 92% of patients in Kenya and 81% in South Africa (SLATE I and II combined) had at least three symptoms of the disease and 67% in Kenya and 63% in South Africa had all four symptoms. Virtually all (96% (84%–99%)) those diagnosed presented with a cough. Weight loss was also common (86% (69%–95%)), followed by night sweats (82% (65%–93%)) and fever (71% (53%–86%)) (table 1).

### TB testing and diagnosis in asymptomatic patients
Among the 337 (63% (58%–67%)) patients with no TB symptoms in SLATE I, 4 (3% (1%–6%)) patients in Kenya and 33 (17% (12%–23%)) patients in South Africa were tested for TB by the study clinics. In Kenya, three of the patients had documented reasons for being tested by the clinic (one was taking cough syrup for the previous 5 days and was recently screened for TB, one had suspected extra-pulmonary TB due to swelling in the jaw and one patient was asthmatic). The remaining patient had no documented reason for a TB test. In South Africa, one of the clinics in our study was attempting to collect sputum from all PLHIV prior to ART initiation, regardless of symptoms. None of the 33 asymptomatic patients tested was positive (figures 1, 2).

In SLATE II, as per the study protocol,[8] all 296 intervention arm patients were asked for a sputum sample. We were able to successfully collect and test sputum for 118 (76% (68%–93%)) asymptomatic patients (figure 3). One positive TB test result was recorded among those who were eligible for SDI under the SLATE II algorithm. This was an asymptomatic patient who had a negative LAM test and CD4 count of 350 cells/mm$^3$. The patient was successfully traced and commenced TB treatment the following day.

### TB LAM test results
In SLATE II, all intervention arm subjects with ≥1 symptoms of TB had a LAM test performed. Only two tests (<1%) were positive. Both patients had sputum samples taken for Xpert testing. The first patient was a man who came to the clinic for care because he felt unwell. He reported all four TB symptoms and had a CD4 count of 78 cells/mm$^3$. This patient's result came back positive for TB and he was successfully traced and commenced on TB treatment. The second patient was a woman who

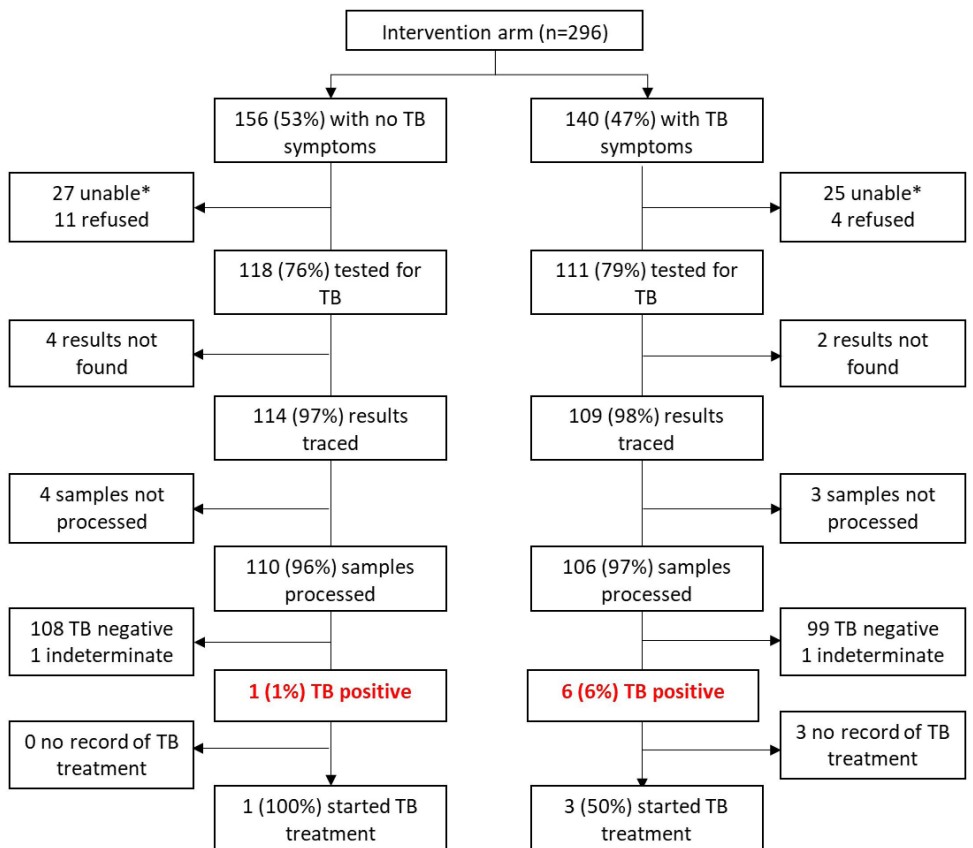

**Figure 3** TB testing flowchart at study enrolment among 296 SLATE II intervention arm participants South Africa. SLATE, Simplified Algorithm for Treatment Eligibility; TB, tuberculosis.

came for an HIV test and reported cough and weight loss and had a CD4 count of 19 cells/mm$^3$. This patient tested negative for TB on Xpert.

### TB treatment uptake
In Kenya, 8 of the 12 patients (67% (38%–88%)) who tested positive for TB went on to initiate treatment for the disease, all within 2 weeks of diagnosis. Of the remaining four patients, one died within 3 weeks of study enrolment with no record of starting TB treatment, and the remaining three remained in care but had no record of initiating TB treatment. Of the 16 patients (symptomatic and asymptomatic) who tested positive in SLATE I and II in South Africa, 8 initiated TB treatment after diagnosis (4 patients within 1 week and 4 within 7 weeks after study enrolment), 4 patients remained in care but had no record of starting TB treatment, 3 patients were transferred to another facility before starting TB treatment and 1 patient was lost from care before starting treatment.

### Unmasking TB disease
One of the major concerns about SDI is immune reconstitution inflammatory syndrome among patients with undiagnosed TB (TB-IRIS). No patients in either study (both standard of care and intervention arms) had indications in their routine clinic records of IRIS reactions during passive study follow-up, though this may reflect incomplete record keeping. Patient clinical records indicated

that clinic staff investigated a total of 25 patients for TB more than 30 days after study enrolment. Of these, 24 test results were available (we were unable to locate one test result in SLATE I in South Africa) and 2 were positive for TB disease (one positive patient was in the standard of care arm in South Africa (SLATE I) and one was in the intervention arm in Kenya (SLATE I) and not eligible for SDI). Among these 24 patients with a follow-up TB test, 4 (17%) (three in SLATE I and one in SLATE II) were eligible for SDI via the SLATE algorithm, none of these patients had a positive TB test.

In the rest of the Results section, we will focus solely on SLATE II in South Africa, where the new TB module in the SLATE II algorithm was applied.

### TB in symptomatic patients eligible for SDI under SLATE II algorithm
In the SLATE II study, patients with milder TB symptoms and a negative TB LAM test were eligible for SDI under the study at the discretion of the study nurse. Of the intervention arm patients who screened in and were eligible for SDI, 40% (34%–46%) (n=101) presented with at least one of the four TB symptoms. Of these, 76% (66%–83%) (n=77) reported only one symptom, 20% (13%–28%) (n=20) reported two symptoms and 4% (1%–9%) (n=4) reported three or more symptoms. The most frequently reported symptoms were weight loss (64%) and cough

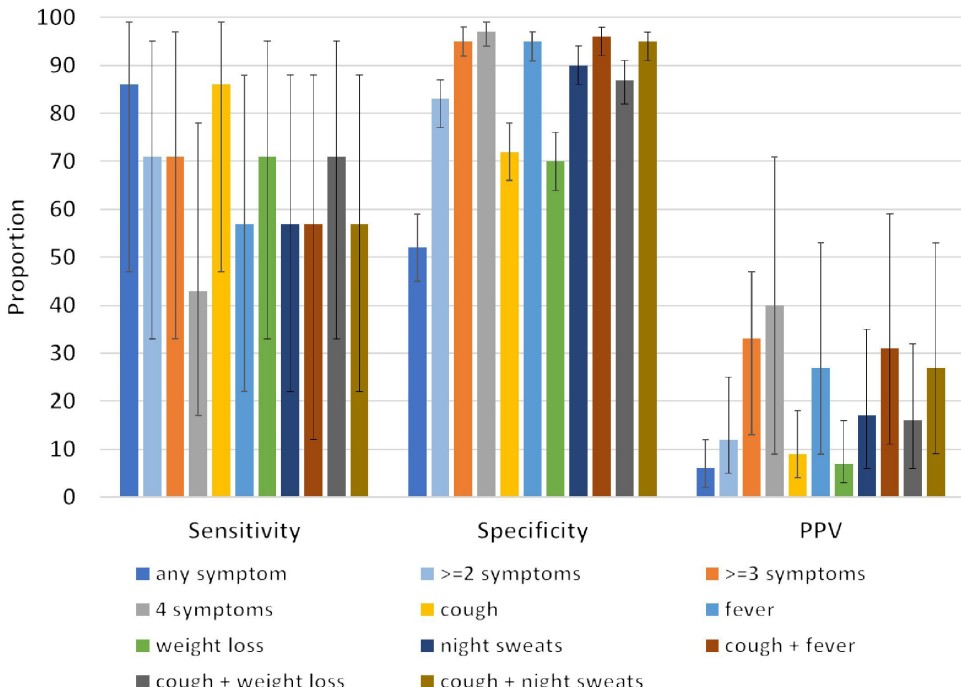

**Figure 4** Sensitivity, specificity and positive predictive value (PPV) (with 95% CIs) for the whom four-symptom screen using Simplified Algorithm for Treatment Eligibility II data.

(45%); night sweats (13%) and fever (6%) were less frequently reported. Of the 101 symptomatic patients, 75% (64%–81%) (n=74) successfully completed a TB test (the remainder either could not produce a sputum sample or were unwilling to do so) and of these, only one tested positive for TB. This patient reported only cough and had a negative LAM test during administration of the SLATE II algorithm. After SDI, the patient, whose CD4 count was 116 cells/mm$^3$, could not be traced to start TB treatment, despite phone calls and a home visit. The patient never returned to the clinic after study enrolment for either TB or HIV care.

### Sensitivity, specificity and predictive values of SLATE II algorithm

Figure 4 displays the results of the performance of the WHO four-symptom screen in SLATE II intervention arm patients. The sensitivity and specificity of any single TB symptom were 86% (47%–99%) and 52% (45%–59%), respectively. The positive predictive value was 6% (2%–12%), while the negative predictive value was 99% (96%–99%). When modifying the definition of a positive TB screen to ≥3 symptoms, sensitivity decreased to 71% (33%–95%) and specificity increased to 95% (92%–98%). The positive predictive value improved to 33% (13%–59%) while negative predictive value remained unchanged (99% (97%–99%)). The WHO symptom screening tool appeared to perform the best for sensitivity (86% (47%–99%)) and specificity (72% (66%–78%)) when using cough only. However, the positive predictive value was the highest in the analysis using ≥3 (33% (13%–59%)) and four (40% (14%–71%)) symptoms. In all other analyses of the WHO four-symptom

screen, sensitivity falls and specificity increases as a result, while the positive predictive value ranges from 7% to 31% depending on how the prevalence shifts. The negative predictive value remained unchanged for all analyses at above 98%. This was expected as the prevalence of TB in the population was low (3%) based on Xpert testing.

### DISCUSSION

In three cohorts of PLHIV not yet on ART and presenting for care in Kenya and South Africa since 2017, we estimated an overall prevalence of any TB symptoms of 35%–50%. Between 60% and 80% of these patients were tested for TB, among whom 19% in Kenya and 6%–15% in South Africa had a positive TB test using Xpert MTB/RIF, corresponding to an estimated prevalence in patients presenting for care of 5% in Kenya and 2%–3% in South Africa (assuming those not tested are negative for TB disease). The results we report for TB symptom prevalence (with cough also being the most common symptom reported) and percentage of patients with TB symptoms tested for TB reported in our study are consistent with what has been previously published at a national level in Kenya[13] and in HIV study cohorts in South Africa.[14–21] The consistency of our estimates of TB disease prevalence with previously published literature varied. Some studies report estimates that were comparable,[13 14] lower[15] or higher.[16–21] National estimates of TB prevalence are still sparse in South Africa. However, the National Tuberculosis Prevalence Survey is currently underway is currently underway in South Africa, so better estimates will be available soon.[22]

In our studies, over 80% of the 28 patients who tested positive for TB disease reported having at least three clinical symptoms of the disease, the most common being a cough, followed by weight loss, night sweats and fever. Other studies have estimated more than 80% of patients diagnosed with TB had more than one symptom of the condition.[18 19 23] A study in South Africa also reported that more than 60% of patients without TB disease had symptoms,[23] again consistent with our findings of 59%.

Using SLATE II results, the performance of the WHO four-symptom screen tool, when classifying patients with ≥1 symptoms of TB, was comparable regarding sensitivity to what was reported in a recent meta-analysis,[4] but we had a higher specificity at 52% (vs 28%[4]). When we increased the number of symptoms required for a positive screen to three or four, or to just four, we saw a decrease in sensitivity and subsequent increase in specificity and an increase in the positive predictive value from 6% (for ≥1 symptoms of TB) to 33% and 40%, respectively.

When delaying ART in person living with HIV is contingent on a positive TB test, the positive predictive value is more clinically relevant than sensitivity or specificity. The main reason for the delay of ART initiation in TB suspect patients is to prevent TB-IRIS. The estimated risk of TB-IRIS is quite low, however, at roughly 1%–6% in sub-Saharan Africa.[24] This risk is highest in patients presenting with CD4 counts <100 cells/mm$^3$.[25 26] The median CD4 count in all three of our cohorts among symptomatic patients was above 100 cells/mm$^3$, while patients presenting with no symptoms had a median CD4 count above 300 cells/mm$^3$. The lowest median CD4 counts, and those at highest risk of TB-IRIS, were among patients who had three if not all four symptoms of TB disease and had a positive Xpert test. In a healthcare system that can produce TB test results soon after ART initiation and successfully contact (trace) patients with positive results, it may be reasonable to increase the number of TB symptoms required to trigger a delay in ART initiation to three or four. This would increase the probability that a patient who is classified as TB positive (using the WHO four-symptom screen) truly has TB disease and allow patients with fewer or milder TB symptoms to start ART while TB test results are pending. It is still unknown whether initiating ART prior to TB being confirmed or treated causes harm (or benefit). In a setting in which TB tests are delayed and/or active tracing of patients with positive results is poor, in contrast, a more conservative approach—delaying ART until a TB test can be completed for patients with even one symptom—may continue to be justified. However, more research is required to accurately weigh the risks of delaying ART (which might prevent a person from being successfully linked into HIV treatment) versus the benefits of waiting for a definitive TB diagnosis to minimise the already low and potentially manageable clinical risk of TB-IRIS.

In both Kenya and South Africa, national TB guidelines during the study stated that PLHIV self-reporting any TB-related symptoms should be tested with Xpert MTB/RIF and treated if diagnosed with the disease. In SLATE I, we saw gaps in the following national guidelines in both countries. Twenty-seven symptomatic, intervention arm patients in Kenya and 45 in South Africa were ineligible for SDI due to TB symptoms and were referred back to the clinic for further testing but were not tested for TB by the clinic staff. We assume that a certain number of patients refused or were unable to provide a sputum sample for testing, but for some, the nurse or clinical officer who saw the patient chose not to do a test. At one study site in South Africa, we were told, informally, that staff only requested TB tests if two or more symptoms were present, while at one site in Kenya, a clinical officer would diagnose a respiratory infection before TB and require the patient to go through a course of antibiotics, advising the patient to return for a TB test only if symptoms persisted. Whether failure to follow guidelines precisely reflects reasonable use of clinical judgement, lack of available resources or irresponsible non-compliance on the part of clinic staff is unclear, but it should be considered in efforts to improve treatment of TB disease and maximise opportunities to offer TB preventative treatment for patients without TB.

The urine TB-LAM test used in SLATE II (the Determine TB LAM Ag, as specified above) was positive in <1% of our intervention arm patients in the SLATE II trial. This test has been shown to improve TB diagnosis in patients with low CD4 counts,[27] so the low yield we saw could be due to higher CD4 counts in our cohort. The average CD4 count in SLATE II intervention arm patients was 294 cells/mm$^3$, it was 175 cells/mm$^3$ among those with TB symptoms and ranged from 60 cells/mm$^3$ to 107 cells/mm$^3$ among those diagnosed with TB. Also, previous studies have shown that the positive predictive value of urine LAM testing depends on the prevalence of TB disease in the population, which was low (6%) in our SLATE II cohort.[28] The LAM test used in the study may thus not offer much benefit for the cost involved.

Our study had several limitations. First, while the study sites were all typical primary healthcare clinics in South Africa and typical hospital-based HIV clinics in Kenya, they were geographically clustered in each country, making generalisability to the rest of the country uncertain. Second, we relied heavily on routinely collected data pertaining to TB test conduct and results, and it is likely that some TB tests were ordered but not analysed or analysed but not recorded. Similarly, data on events after the study enrolment visit, such as postinitiation TB diagnoses or disease, were likely incomplete, and some events may not have been reported to healthcare facilities at all. Third, we assumed that patients who did not have TB tests were TB-negative; it is possible that some in fact had TB, making prevalence in our population higher than reported. Finally, our sample sizes were small, limiting our ability to stratify by patient characteristics; a larger sample might identify other characteristics associated with a positive TB test. The small sample size could also affect our ability to detect TB-related adverse events in our sample.

Despite these limitations, we conclude that in the SLATE I and SLATE II trials, among 235 patients with WHO-defined TB symptoms who were not eligible for SDI, over 80% did not have TB, making any delay to ART initiation due to the requirement for a preinitiation TB test unnecessary. No serious, TB-related adverse events were reported after starting ART among symptomatic patients with or without delay in our study. Reconsideration of WHO's guidance to even 'briefly' delay ART initiation based on the presence of 'any TB symptom' may be appropriate, with ART initiation among patients with fewer or milder symptoms commencing while TB test results are still pending.

**Author affiliations**
[1]Departments of Epidemiology, Boston University, Boston, Massachusetts, USA
[2]Health Economics and Epidemiology Research Office, Department of Internal Medicine, School of Clinical Medicine, Faculty of Health Sciences, University of the Witwatersrand, Johannesburg, South Africa
[3]Department of Global Health, Boston University, Boston, Massachusetts, USA
[4]Kenya Medical Research Institute, Nairobi, Kenya
[5]Henry M. Jackson Foundation Medical Research International, Inc, Nairobi, Kenya
[6]Ezintsha, Faculty of Health Sciences, University of the Witwatersrand, Johannesburg, South Africa
[7]Bill & Melinda Gates Foundation, Seattle, Washington, USA

**Acknowledgements** We thank the patients who participated in the study and the staff of the six study clinics for their cooperation; the City of Johannesburg and Ekhureleni Metro in South Africa and the management of Kericho County Referral Hospital, Kapsabet County Referral Hospital and Kombewa County Hospital in Kenya for permission to conduct this study.

**Contributors** MF, WDFV and SR had substantial contributions to the conception and design of the work. AB, MM and BAL acquisition, analysis or interpretation of data. AB, MM, BAL, IT, MB, LV, MF, WDFV, PDE and SR contributed substantially to the drafting and revising the work critically for important intellectual content. All the authors have read and approved the final version of the manuscript. All authors are willing to be accountable for the work and in ensuring that questions related to the accuracy and integrity of the work were appropriately investigated and resolved.

**Funding** Research was funded by the Bill and Melinda Gates Foundation (Grant number OPP1136158).

**Competing interests** None declared.

**Patient consent for publication** Not required.

**Ethics approval** Approval of the SLATE trials was provided by the Human Research Ethics Committee of the University of the Witwatersrand in South Africa (SLATE I 160910; SLATE II 171011), the Kenya Medical Research Institute (SLATE I 3408) and the Walter Reed Army Institute of Research in Kenya (SLATE I 2401) and the Institutional Review Board of Boston University (SLATE I H-35634; SLATE II H-37010).

**Provenance and peer review** Not commissioned; externally peer reviewed.

**Data availability statement** Data generated by the study will be made publicly available in the Dryad repository (http://www.datadryad.org/) after the protocol has been closed (anticipated closure December 2020). Until then, data will remain under the supervision of the Boston University Medical Campus IRB and the University of the Witwatersrand Human Research Ethics Committee (HREC). Requests can be sent to the BUMC IRB at medirb@bu.edu. Data extracted from routine medical records are owned by the study sites and the South African National Department of Health and cannot be made publicly available by the authors.

**ORCID iDs**
Alana Brennan http://orcid.org/0000-0002-7746-0668
Bruce A Larson http://orcid.org/0000-0002-9322-2387
Willem Daniel Francois Venter http://orcid.org/0000-0002-4157-732X
Sydney Rosen http://orcid.org/0000-0002-6560-2964

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
