## [Reviewer comments · BMJ Open]

ARTICLE DETAILS

TITLE (PROVISIONAL)	Prevalence of TB symptoms, diagnosis and treatment among people living with HIV (PLHIV) not on ART presenting at outpatient clinics in South Africa and Kenya: baseline results from a clinical trial
AUTHORS	Brennan, Alana; Maskew, Mhairi; Larson, Bruce; Tsikhutsu, Isaac; Bii, Margaret; Vezi, Lungisile; Fox, Matthew; Venter, Willem Daniel Francois; Ehrenkranz, Peter; Rosen, Sydney

VERSION 1 – REVIEW

REVIEWER	Eveline Klinkenberg independent consultant
REVIEW RETURNED	25-Dec-2019

GENERAL COMMENTS	nicely written paper but still needs some attention before it can be published. Key overall points: 1) Paper needs careful reading before submitting as there are several repeat words, words missing, please do a final check before final submission2) The main issue with the paper is that the aim is not well enough worded in the abstract, see detailed input in the uploaded file3) Consider not distinguishing between SLATE I and II in the abstract but just state SLATE unless the trial including the arms are better explained within the word space. A good intro of the SLATE trial would anyhow be required in the abstract for the reader to understand.4) some additional edits are in the attached uploaded file The reviewer provided a marked copy with additional comments. Please contact the publisher for full details.
--

REVIEWER	Khaing Hnin Phyo International Union Against Tuberculosis and Lung Disease (The Union), Myanmar
REVIEW RETURNED	23-Apr-2020

GENERAL COMMENTS	Thank you for the author interest in TB HIV coinfection. This study was good. However, please kindly see the following comments: 1. For the overall manuscript - it is better to describe as People Living with HIV (PLHIV) not on ART rather than HIV-infected or HIV-positive patients. Recommended to check some English wording and rephrase accordingly for easy understanding for the reader.2. In the abstract - The result session, sentence number 25; 'When comparing.....value to >30%' is needed to rephrase for easy
--

	understanding for the reader. Delayed ART initiation is needed to mention in the result session. 3. In method session of the manuscript - the four TB symptoms is repeated frequently and in my opinion, the details of TB symptoms can only be described once and 'TB symptoms' can be used. 4. In discussion session - It was written fine. The limitation was also stated clearly.
--	--

VERSION 1 – AUTHOR RESPONSE

Reviewer 1 (Eveline Klinkberg)

Nicely written paper but still needs some attention before it can be published. Key overall points:

1. *Paper needs careful reading before submitting as there are several repeat words, words missing, please do a final check before final submission.*

Thank you. We apologize for these mistakes and have carefully re-edited the manuscript.

2. *The main issue with the paper is that the aim is not well enough worded in the abstract, see detailed input in the uploaded file.*

Thank you. We address this and all concerns/comments in the uploaded file in our responses below.

3. *Consider not distinguishing between SLATE I and II in the abstract but just state SLATE unless the trial including the arms are better explained within the word space. A good intro of the SLATE trial would anyhow be required in the abstract for the reader to understand.*

We thank the reviewer for their comment. Within the word constraint for the abstract, we have tried to introduce SLATE I and SLATE II to allow clear presentation of results by study and country. We note that this manuscript focuses on baseline data only; detailed knowledge of the trials themselves is not needed to understand this manuscript. Since protocols and primary outcomes for both trials have already been published and/or presented, more information about the trials risks simply lengthening this manuscript without improving readers' understanding.

4. *Some additional edits are in the attached uploaded file.*

We list the comments from the uploaded file here and explain how we have addressed them. The page number is the manuscript page number (x of 28) imbedded in the reviewer's pdf document (not the pdf page number).

- *Page 1, Line 3: word missing in the title*

We have revised accordingly.

- *Page 2, Line 4: needs more clear intro that aim is to check those not initiated on ART whether they have TB or not to link it to the conclusions drawn.*

We have added the following sentence to the Abstract to try to clarify our aim: "One of the major barriers to same-day initiation has traditionally been the need to rule out tuberculosis (TB) prior to starting ART, leading to initiation delays for patients with any TB symptoms."

- *Page 2, Line 7 data from 2 countries only... can you claim this is representative for SSA?!?*

No, we definitely did not intend to claim our results are representative of SSA. We have changed the reference to sub-Saharan Africa in the Abstract to "South Africa and Kenya" to avoid any chance of

this interpretation. One of the limitations we include is that geographic generalizability is limited, even within the study countries. This is also mentioned in the “strengths and limitations” box.

- *Page 2, Line 19 if results in the abstract are split in SLATE I and II this needs introduction as to what is the difference in these arms*

We revised the abstract to explain that the studies evaluated different algorithms for same-day initiation (SLATE I and SLATE II). We are unable to go into further details about the differences between the two trials due to the word limits of the abstract.

- *Page 2, Line 30: for xxx ART initiation was delayed, also this sentence suggest this was deliberately delayed while the results do not show anything on ART initiation as such so this needs better alignment. Results is for those not on ART but no results are presented as to the why of that...*

We have revised the conclusions of the abstract to be consistent with the main objective of this manuscript. It now reads, “80% of patients assessed for ART initiation presented with >1 TB symptoms. Reconsideration of the “any symptom” rule may be appropriate, with ART initiation among patients with fewer/milder TB symptoms commencing while TB test results are pending.”

- *Page 3, Line 20: so if this is uncertain even at country level you cannot generalize to SSA*

We agree. As explained above, we did not intend to generalize to SSA.

- *Page 5 line 5: or?? what is the aim of this assessment...*

We are sorry for any confusion here. Assessing the performance of any screening test is done by comparing actual test results to the patient's true disease status (as assessed by a gold standard). The four measures used to evaluate the WHO 4 symptom screening test are the sensitivity, specificity, and positive and negative predictive values.

We describe what we did in detail in the methods section on page 8 manuscript, “we calculated the sensitivity (probability of screening positive (using each definition above) when TB disease is present as defined by Xpert® MTB/RIF), specificity (probability of screening negative when TB disease is not present as defined by Xpert® MTB/RIF), positive predictive value (probability of a patient having TB disease when the screen is positive) and negative predictive value (the probability of a patient not having TB disease when the screen is negative).”

- *Page 7 line 44: so the conclusion are only based on this group of participants?! that is not clear from the abstract....*

We have revised the abstract to specify that the conclusion is based on this group of participants.

- *Page 9 line 5: in discussion need to assume what would results look like if not all were TB neg... if this assumption holds also really depends on the effort made to collect sputum from each participant...*

We are not certain that we understand this comment, but we have added the following statement to the paragraph on limitations of the study in the Discussion: “Third, we assumed that patients who did not have TB tests were TB-negative; it is possible that some in fact had TB, making prevalence in our population higher than reported.” The effort made to collect sputum should not affect whether patients were actually positive or negative, but only whether they could be tested.

- *Page 9, line 20: check if in discussion the 30% Lost for testing at the start is taken up as point, this is important!*

The large proportion of symptomatic patients who did not receive TB tests is addressed in the discussion. We state the following, “In SLATE I, we saw gaps in following national guidelines in both

countries. Twenty-seven symptomatic, intervention arm patients in Kenya and 45 in South Africa were ineligible for SDI due to TB symptoms and were referred back to the clinic for further testing but were not tested for TB by the clinic staff. We assume that a certain number of patients refused or were unable to provide a sputum sample for testing, but for some, the nurse or clinical officer who saw the patient chose not to do a test. At one study site in South Africa we were told, informally, that staff only requested TB tests if two or more symptoms were present, while at one site in Kenya, a clinical officer would diagnose a respiratory infection before TB and require the patient to go through a course of antibiotics, advising the patient to return for a TB test only if symptoms persisted.”

- *Page 12, line 44: make clear as part of this study/intervention.; and also important to mention bac confirmed TB in this context*

We have added that the diagnosis was made using Xpert MTB/RIF; we are sorry, but we do not understand the rest of the reviewer’s comment.

- *Page 13: line 5: odd statement, not sparse, not available except based on global WHO estimates but currently national survey has been completed and an evidence based estimate will follow in the near future*

We apologize for not being clear. We have added the following to the manuscript, “However, the National Tuberculosis Prevalence Survey is currently underway, so better estimates will be available soon.”

- *Page 13: Line 16: make clear what the consistency is here. 80% at least one of three clinical symptoms is not the same as multiple symptoms to which reference is made in the next sentence so this part can be made more explicit, what is the point the authors want to make*

We are sorry for the lack of clarity. The point we would like to make is that the prevalence of symptoms in our study was similar to that found in other studies. We feel that “consistent with” the studies we cite is a reasonable description, but we will defer to the editors if there is a better way to phrase this.

- *Page 13 line 25: be explicit: from at least one to >3 or 4? (so >3)*

We apologize for the lack of clarity and have revised this sentence.

Reviewer 2 (Khaing Hnin Phyo)

Thank you for the author interest in TB HIV coinfection. This study was good. However, please kindly see the following comments:

1. *For the overall manuscript - it is better to describe as People Living with HIV (PLHIV) not on ART rather than HIV-infected or HIV-positive patients. Recommended to check some English wording and rephrase accordingly for easy understanding for the reader.*

Thank you for this advice. We have revised the paper throughout to use PLHIV. We have also proofread carefully to improve ease of understanding for the reader.

2. *In the abstract - The result session, sentence number 25; 'When comparing.....value to >30%' is needed to rephrase for easy understanding for the reader. Delayed ART initiation is needed to mention in the result session.*

We have revised the abstract results section to clarify this sentence. We have also revised the conclusions and eliminated the phrase “delayed ART initiation”.

- 3. In method session of the manuscript - the four TB symptoms is repeated frequently and in my opinion, the details of TB symptoms can only be described once and 'TB symptoms' can be used.*

We have edited where possible to incorporate this recommendation.

- 4. In discussion session - It was written fine. The limitation was also stated clearly.*

Thank you.

VERSION 2 – REVIEW

REVIEWER	Khaing Hnin Phyo International Union Against Tuberculosis and Lung Disease, The Union, Myanmar
REVIEW RETURNED	22-Jun-2020
GENERAL COMMENTS	This is an interesting article and I have no specific comments.